# Downsizing of COVID-19 contact tracing in highly immune populations

**Maria M. Martignoni[1]\***, **Josh Renault[2]**, **Joseph Baafi[2]**, **Amy Hurford**[1,2]

**1** Department of Mathematics and Statistics, Memorial University of Newfoundland, St. John's (NL), Canada,
**2** Department of Biology, Memorial University of Newfoundland, St. John's (NL), Canada

\* mmartignonim@mun.ca

**Data Availability Statement:** The computer code is publicly available at https://figshare.com/articles/figure/Code_for_Figures_of_the_manuscript/19103183.

## Abstract

Contact tracing is a key component of successful management of COVID-19. Contacts of infected individuals are asked to quarantine, which can significantly slow down (or prevent) community spread. Contact tracing is particularly effective when infections are detected quickly, when contacts are traced with high probability, when the initial number of cases is low, and when social distancing and border restrictions are in place. However, the magnitude of the individual contribution of these factors in reducing epidemic spread and the impact of population immunity (due to either previous infection or vaccination), in determining contact tracing outputs is not fully understood. We present a delayed differential equation model to investigate how the immunity status and the relaxation of social distancing requirements affect contact tracing practices. We investigate how the minimal contact tracing efficiency required to keep an outbreak under control depends on the contact rate and on the proportion of immune individuals. Additionally, we consider how delays in outbreak detection and increased case importation rates affect the number of contacts to be traced daily. We show that in communities that have reached a certain immunity status, a lower contact tracing efficiency is required to avoid a major outbreak, and delayed outbreak detection and relaxation of border restrictions do not lead to a significantly higher risk of overwhelming contact tracing. We find that investing in testing programs, rather than increasing the contact tracing capacity, has a larger impact in determining whether an outbreak will be controllable. This is because early detection activates contact tracing, which will slow, and eventually reverse exponential growth, while the contact tracing capacity is a threshold that will easily become overwhelmed if exponential growth is not curbed. Finally, we evaluate quarantine effectiveness in relation to the immunity status of the population and for different viral variants. We show that quarantine effectiveness decreases with increasing proportion of immune individuals, and increases in the presence of more transmissible variants. These results suggest that a cost-effective approach is to establish different quarantine rules for immune and nonimmune individuals, where rules should depend on viral transmissibility after vaccination or infection. Altogether, our study provides quantitative information for contact tracing downsizing in vaccinated populations or in populations that have already experienced large community outbreaks, to guide COVID-19 exit strategies.

**Funding:** JR is supported by a National Sciences and Engineering Research Council of Canada (NSERC) Undergraduate Student Research Award (USRA). AH acknowledges financial support from an NSERC Discovery Grant, RGPIN 2014-05413. MM and AH are supported by Canadian Network for Modelling Infectious Diseases - Reseau canadien de modelisation des maladies infectieuses (CANMOD) and the Department of Health and Community Services, Government of Newfoundland and Labrador. AH acknowledges further support from the NSERC Emerging Infectious Disease Modelling Consortium. AH and JB are supported by the Atlantic Association for Research in the Mathematical Sciences and the New Brunswick Health Research Foundation. The funders had no role in study design, data collection and analysis, decision to publish, or preparation of the manuscript.

**Competing interests:** The authors have declared that no competing interests exist.

# 1 Introduction

Contact tracing, in combination with quarantine, is a key component of the successful management of the COVID-19 pandemic. When contact tracing is in place, people with a confirmed infection provide information about individuals they have been in contact with during the previous days, who are in turn at risk of developing an infection. Identified contacts are traced and quarantined, and quick tracing can significantly slow down, or even prevent, epidemic spread, by quarantining infectious individuals before they become contagious. Efficient contact tracing may allow for partial relaxation of social distancing requirements and border restrictions, particularly during vaccine roll-out.

Different studies have focused on understanding the impact of contact tracing practices on COVID-19 outbreaks [1–10]. Successful strategies involve quick detection of infectious cases (for example through testing) [1, 3, 4, 6, 7, 10], a high probability that contacts are traced [2, 5, 7, 10], a low number of initial cases when contact tracing is implemented [2], and, more generally, maintaining social distancing [2, 4–6], where different variants can affect viral transmissibility [11, 12].

The implementation of border control measures can also impact contact tracing management [13]. Case importations contribute to epidemic spread when infection prevalence is low [14], and it is critical to consider how contact tracing practices should respond to the relaxation of border restrictions, in addition to the relaxation of social distancing. However, the link between case importations and the contact tracing efficiency needed to keep an outbreak under control has been little explored [15].

Finally, population immunity (acquired either through natural infection or vaccination) can clearly affect contact tracing management [16–18]. Especially, with vaccines availability, we expect downsizing, or even dismantlement, of contact tracing to be possible, but only few contact tracing models directly account for vaccine roll-out [19, 20]. Additionally, the proportion of immune individuals in a community may affect quarantine effectiveness, as isolation of contacts who have a significantly lower probability to get infected [21, 22], may lead to unnecessary costs related to isolation of healthy individuals [23].

While the factors increasing contact tracing efficiency during the COVID-19 outbreak have been identified, the magnitude of their individual contribution in reducing epidemic spread is not fully understood [9]. Most of the current approaches have been based on stochastic frameworks which, despite accounting for potential sources of uncertainty in disease transmission, limit the derivation of analytical expressions to quantitatively understand the interplay of different interventions during an outbreak [4].

Here, we present a continuous time approach to disentangle and quantify the impact of individuals' immunity and relaxation of social distancing requirements on contact tracing practices. More specifically, we determine the minimal contact tracing efficiency needed to keep an outbreak under control, in relation to the contact rate and to the proportion of immune individuals in a population (section 3.1). Additionally, we quantify the impact of delays and case importations on the epidemic dynamics, to determine whether the contact tracing capacity should be adjusted in immune populations (section 3.2). We also consider how population immunity affects quarantine effectiveness, expressed as the proportion of people that will develop an infection while in quarantine (section 3.3).

Our study provides information about contact tracing downsizing during the relaxation of COVID-19 restrictions in immune populations, and provides insights into the impact of contact tracing policies in different jurisdictions. The model presented here is general, and can be applied to different initial conditions, indicating differences in infection prevalence in a

community. However, our focus will be on jurisdictions that have followed an elimination approach [24], where the infection is introduced in an initially virus-free community.

## 2 Model and methods

To understand the impact of contract tracing on the epidemic we model transmission rates as a product of the average number of contacts, where quarantine is considered in a compartmental model similarly to what done by Lipsitch *et al.* [25], and later followed by several others (e.g., [26–28]). One of the factors that makes COVID-19 particularly difficult to trace is the abundance of asymptomatic individuals in the population, who are spreading the disease without awareness of their infectious status [29]. Thus, in our model we differentiate infectious cases into three subclasses, namely pre-symptomatic, symptomatic, and asymptomatic (analogously to [16]). While in [25] individuals are quarantined based on contact with infected individuals, here we assume that individuals are quarantined based on contact with symptomatic individuals only. Indeed, individuals developing symptoms are more likely to seek testing and become aware of their infectious status. We consider asymptomatic and pre-symptomatic individuals unlikely to seek testing, especially as we consider the situation where the community is not experiencing any outbreak prior to disease introduction.

The pre-symptomatic status lasts on average 2–3 days [16], and if we assume that symptomatic individuals are tested and begin contact tracing 1–2 days after symptoms onset, then individuals can be contagious for five days before their contacts are traced and quarantined. To account for this delay, the contact tracing dynamics is formulated according to a system of delayed differential equations. A detailed description of model dynamics is provided in the Appenix, while a schematic representation of the model compartments is provided in Fig 1. To understand the impact of different interventions and contact tracing practices, we consider the Susceptible-Exposed-Asymptomatic-Infected-Recovered (SEAIR) framework developed by Miller *et al.* [16] (described in Appendix A.1 in S1 File) and extend it to incorporate contact tracing (A.2), immunity status (A.3), delays in outbreak detection (A.4), case importations (A.5), and quarantine effectiveness (A.6).

Model parameters are given in Table 1. Parameters are based on the population size and public health regulations of the Canadian province of Newfoundland and Labrador, as this province lends itself to this study for having followed an elimination approach to reduce COVID-19 transmission, and having implemented extensive contact tracing and quarantine measures to end transmission whenever an outbreak has occurred.

## 3 Results

### 3.1 Contact tracing efficiency and outbreak control

We define the contact tracing efficiency $q$ as the proportion of symptomatic individuals whose contacts will be traced, multiplied by the proportion of contacts that will be quarantined. When the number of cases is low, we can derive an expression for the minimal contact tracing efficiency needed to avoid a growth in the number of cases (see Appendix B, section B.1 in S1 File). We find that disease spread does not occur as long as:

$$q \geq \frac{\alpha c \tilde{b} \frac{S_0}{N} - \tilde{\delta}}{p_{I_c} \alpha c (2b_c + 3b_p) \frac{S_0}{N} + p_{I_c}} , \tag{1}$$

where $S_0/N$ represents the proportion of initially susceptible individuals; $c$ is the contact rate; $\alpha$ is the probability of infection given a contact, $\tilde{b}$ is defined as the weighted average of the

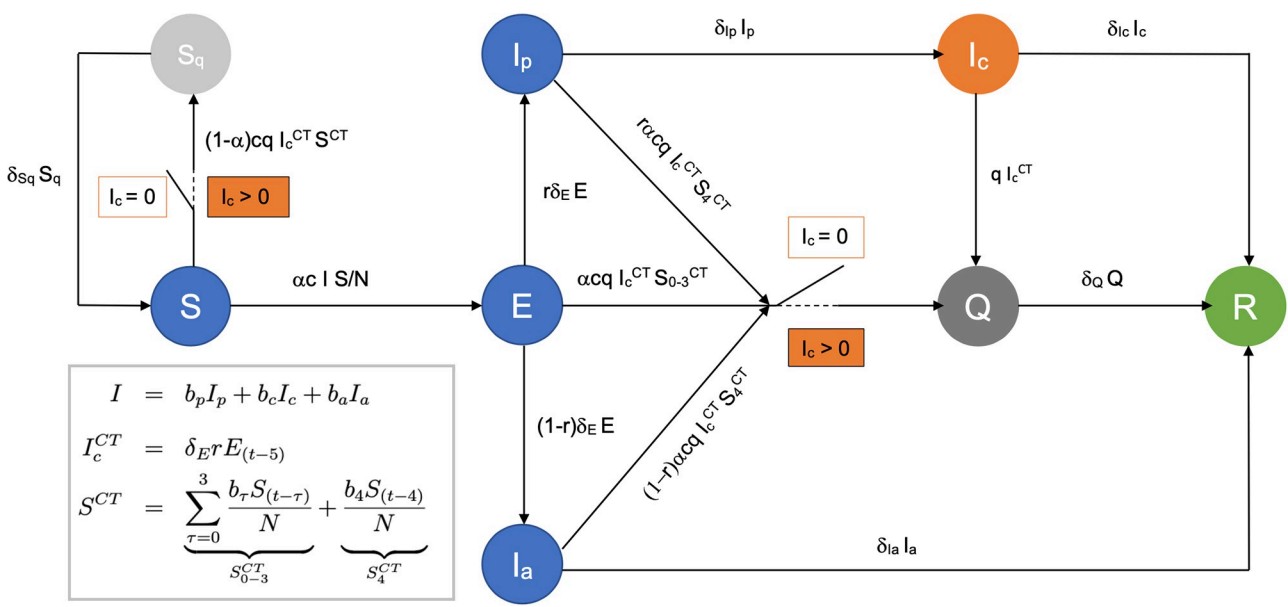

**Fig 1. A schematic representation of the flow of individuals among different compartments as described by the system of Eq (11).** Model parameters are described in Table 1. Susceptible individuals $S$ can become exposed $E$ after having interacted with an infected person, and successively become infectious pre-symptomatic $I_p$, symptomatic $I_c$, or asymptomatic $I_a$. Individuals in states represented in blue are not aware of their infectious status and behave normally in the population. Individuals in $I_c$, represented in orange, can become aware of their status and be contact traced, where $I_c^{CT}$ represents the number of people whose contacts are being traced today (see Eq (6)) and $S^{CT}$ is the total number of identified contacts that each contact tracing individuals has had in the past five days (see Eq (7)). Once contact tracing is activated (for $I_c > 0$, see Eq (5)) contacts can be moved to quarantine (states in grey), where contacts developing an infection will enter the $Q$ class, and contacts not developing an infection will enter the $S_q$ class. Recovered individuals enter the $R$ status (in green). Delays in outbreak detection are modelled by assuming that contact tracing activates only when $I_c > I_c^*$, rather than when $I_c > 0$ (see Appendix A.4 in S1 File).

adjusted contact rates of pre-symptomatic ($b_p$); symptomatic ($b_c$) and asymptomatic individuals ($b_a$); $\tilde{\delta}$ is the average length of the infectious status $\tilde{\delta}$; and $p_{I_c}$ is the time-dependent proportion of symptomatic individuals. Equality in Eq (1) is obtained when $q = q_0^*$, where $q_0^*$ is the minimal contact tracing efficiency required to avoid epidemic spread. A graphical representation of $q_0^*$ as a function of the proportion of immune individuals in a population and of the contact rate is provided in Appendix B, Fig B1 in S1 File.

When the contact rate $c$ is high, and $\alpha c S_0/N$ is relatively big with respect to $\tilde{\delta}$ and $p_{I_c}$ (i.e., when proportion of immune individuals is low and the proportion of symptomatic infections is high), $q_0^*$ approaches a constant value, namely:

$$q_0^* \simeq \frac{\tilde{b}}{p_{I_c}(2b_c + 3b_p)} \, , \tag{2}$$

with $\tilde{b}/(2b_c + 3b_p) < 1$. For $0 \leq q_0^* < 1$ there exists a minimal contact tracing efficiency that can prevent infection spread, even in the absence of social distancing requirements, and in nonimmune populations (e.g., in schools or other unvaccinated settings). Thus, as long as $q \geq q_0^*$ (as it is the case for the parameter space considered), contact tracing can be considered an effective sole intervention against COVID-19.

Our simulations confirm the relationship found in Eq (20). In Fig 2 the minimal contact tracing efficiency $q_c^*$ needed to avoid overwhelming contact tracing capacity is computed as a function of the proportion of immune individuals in a population and of the contact rate. We

**Table 1. Brief description of the variables and parameters of the model given in Eq (11), with corresponding default values or ranges considered for the simulations.** Values in brackets correspond to the explored ranges.

| Variable/parameter | Description | Value/Range |
|---|---|---|
| $S$ | Susceptible population | – |
| $E$ | Exposed population | – |
| $I_p$ | Pre-symptomatic infected population | – |
| $I_c$ | Symptomatic infected population | – |
| $I_a$ | Asymptomatic infected population | – |
| $Q$ | Infected population in quarantine | – |
| $S_q$ | Not-infected population in quarantine | – |
| $R$ | Recovered population | – |
| $N$ | Total population | 500,000 [**] |
| $\alpha$ | Probability of infection given a contact | 0.2 (0.1–0.5) [39] |
| $c$ | Contact rate | 5 (3–9) [40] |
| $b_p$ | Standard contact rate (pre-symptomatic) | 1 |
| $b_c$ | Reduction in contact rate (symptomatic) | 0.75 [*] |
| $b_a$ | Reduction in infectiousness (asymptomatic) | 0.5 [41] |
| $q$ | Contact tracing efficiency | 0.75 (0–1) |
| $r$ | Probability of becoming symptomatic given infected | 0.7 [29] |
| $I_{CT}^{max}$ | Contact tracing capacity | 250 (125, 500) [**] |
| $I_c^*$ | Symptomatic population when contact tracing is activated | 0 (0–9) |
| $\delta_E$ | Rate of people leaving state $E$ daily | 1/4 [16] |
| $\delta_{I_p}$ | Rate of people leaving state $I_p$ daily | 1/2.4 [16] |
| $\delta_{I_c}$ | Rate of people leaving state $I_c$ daily | 1/3.2 [16] |
| $\delta_{I_a}$ | Rate of people leaving state $I_a$ daily | 1/7 [16] |
| $\delta_Q$ | Rate of people leaving state $Q$ daily | 1/14 [**] |
| $\delta_{S_q}$ | Rate of people leaving state $S_q$ daily | 1/14 [**] |
| $p_v$ | Proportion of immune individuals in the population | 0–1 |
| $T$ | Total simulation time | 180 [days] |

[*] Estimated parameters

[**] Parameters based on public health regulations of Newfoundland and Labrador.

find that, for the parameter space considered, when the contact tracing efficiency $q$ is large enough (i.e., $q > 0.5$ in Fig 2), overwhelming contact tracing capacity does not occur, independently from the contact rate and immunity status of the population, and contact tracing alone is a sufficient measure to keep epidemic spread under control. We see, for example, that the outbreak can be controlled without contact tracing if 55% of the population is fully immune when the contact rate $c = 3$, or if 85% of the population is fully immune when the contact rate $c = 7$. Alternately, in non immune populations, the outbreak can be controlled with contact tracing efficiency $q > 0.4$ when the contact rate is 3, and with $q > 0.5$ if the contact rate $c = 7$. Different variants (expressed as differences in the probability of infection given a contact $\alpha$) can also affect the minimal contact tracing efficiency needed to avoid overwhelming contact tracing capacity, where higher contact tracing efficiency is needed to control outbreaks of more contagious variants.

We found that the contact tracing capacity $I_{CT}^{max}$ does not significantly affect the results (Fig B2 in S1 File). In a parameter space where the outbreak can be controlled, disease spread, and consequently overwhelming contact tracing capacity, will not occur. If the outbreak can not be

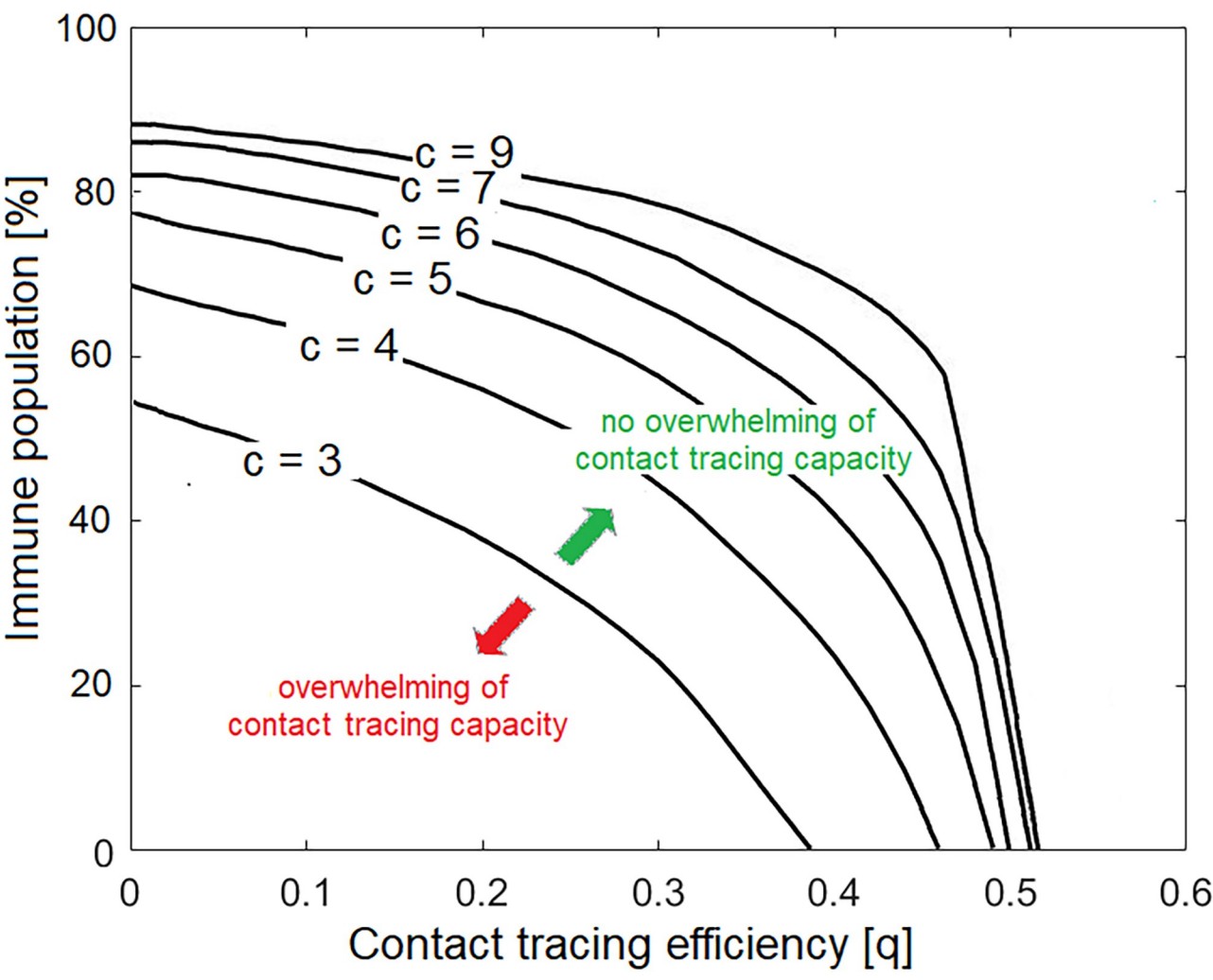

**Fig 2. Contact tracing efficiency needed to avoid overwhelming contact tracing capacity, as a function of the proportion of immune individuals in a population and for different contact rates (black curves, with $c$ = {3, 4, 5, 6, 7, 9}), for $I_c^*$=0.** The area below each curve represents the parameter space for which contact tracing is overwhelmed, while the area above each curve represents the parameter space for which contact tracing is not overwhelmed. The curves represents the minimal contact tracing efficiency $q_c^*$ needed to avoid contact tracing overwhelming. Default parameters used for the simulations are given in Table 1.

controlled, the number of cases will grow nearly exponentially and exceed the contact tracing capacity $I_{CT}^{max}$, where doubling or halving $I_{CT}^{max}$ will not significantly affect the results. Therefore, the minimal contact tracing efficiency needed to avoid overwhelming contact tracing ($q_c^*$, obtained numerically) and the minimal contact tracing efficiency needed to avoid a growth in the number of cases ($q_0^*$, obtained analytically in Eq (20)), appear to be similar quantities (cfr. Fig 2 and B2 with Fig B1 in S1 File). Efficiency, in terms of quick detection of symptomatic cases and identification and quarantining of their contacts, and not contact tracing capacity, is therefore the most important determinant of whether an outbreak will be controlled or not. Highly efficient contact tracing should keep the number of infections, and thus the number of contact tracing individuals, below capacity.

### 3.2 Delayed detection and case importations

We consider the situation in which contact tracing is efficient enough to control the outbreak (i.e., $q = 0.75$, cfr. Fig 2) and investigate the impact of delayed detection on contact tracing capacity, where delays are modeled as an increase in the number of symptomatic cases present in the community when contact tracing is activated (i.e., $I_c^*$, see Eq (13)). In Fig 3a–3c we see that if a delay in detection is experienced, the number of cases to be traced per day increases, particularly when the proportion of immune individuals in a population is low. An increase in the daily number of cases can lead to a non-controllable outbreak, even when the contact tracing efficiency is high. For example, we see that if the outbreak is detected only when already 6 symptomatic cases are present (see. Fig 3b), in nonimmune populations and with a contact rate of 7, the number of contacts to be traced is around 400 a day. High immunity however, minimizes the impact of delays, and helps to maintain contact tracing within capacity, even when the contact rate is high. For example, if 60% of the population is immune and the outbreak is detected when 6 symptomatic cases are already in the community, the maximum number of contacts to trace with a high contact rate of 7 per day is around 50, and thus more easily manageable (Fig 3b).

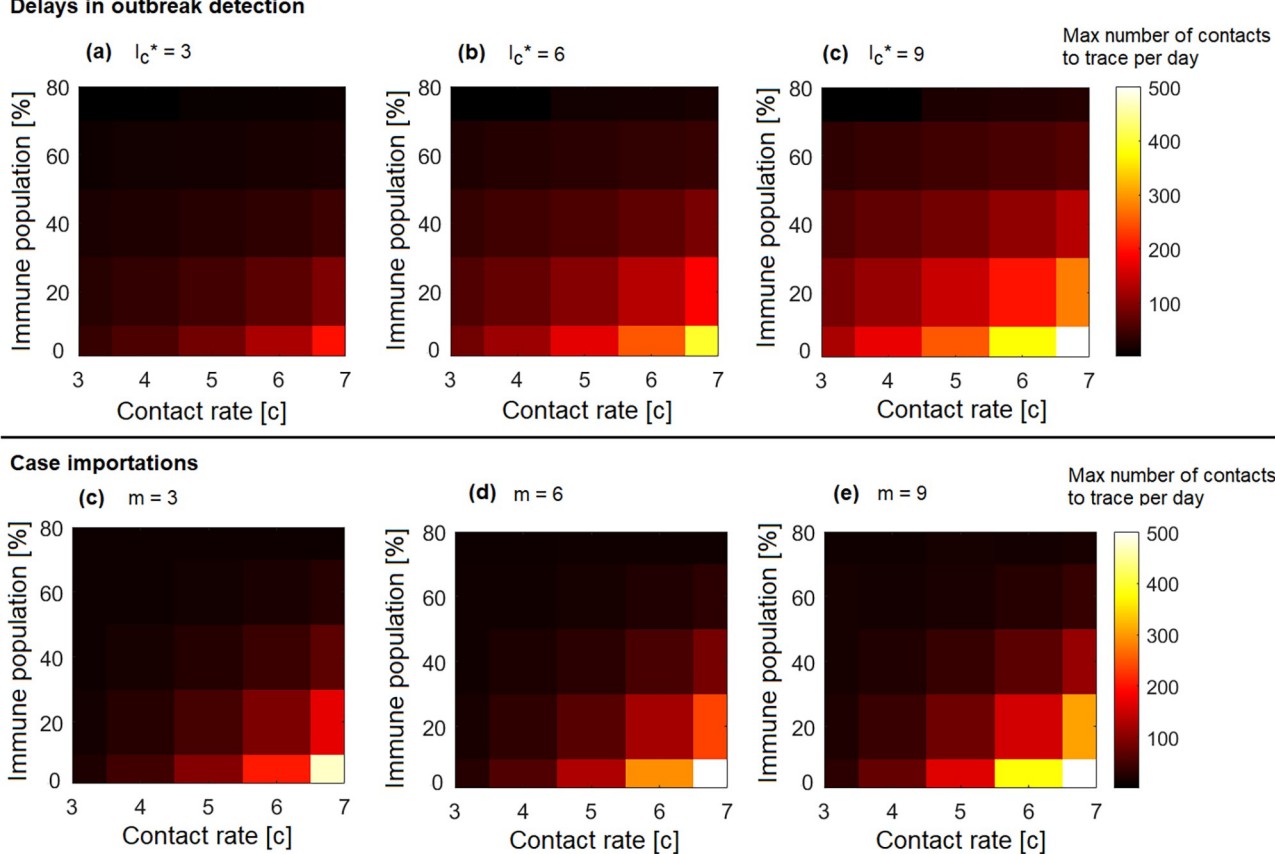

**Fig 3. Maximal number of contacts to trace per day as a function of the proportion of immune individuals in the population and the contact rate.** The number of symptomatic individuals already present in the community when contact tracing is activated ($I_c^*$) is progressively increased from (a) $I_c^* = 3$, to (b) $I_c^* = 6$, to (c) $I_c^* = 9$. In figures (d)-(f) $I_c^* = 3$ for all simulations, and the number of imported cases $m$ over the time interval considered (i.e., $T = 180$ days) varies, with (d) $m = 3$, (e) $m = 6$, and (f) $m = 9$. The contact tracing efficiency is kept constant at $q = 0.75$.

Case importations can also lead to overwhelming contact tracing capacity (Fig 3d and 3e), however in immune populations this risk is strongly reduced. Indeed, if for example a number of 6 imports are experienced in 180 days, the number of contacts to trace per day might reach 500 in the absence of immunity, while it remains lower than 40 when 60% of the population is immune (Fig 3e). Note however that when a region experiences multiple imports, each import incorporates a risk for delayed detection, leading to an increased risk of exceeding capacity (as seen in Fig 3a–3c). Swift detection of infected imported cases is therefore important to make sure that the contact tracing capacity is not overwhelmed. Additionally, in the simulations we assumed imports to be evenly distributed over the time interval considered. However, imports might occur simultaneously, which could increase the risk of overwhelming contact tracing capacity.

### 3.3 Quarantine effectiveness

We look at how quarantine effectiveness, intended as the proportion of quarantined individuals that develop an infection, is affected by the proportion of immune individuals in a population and by disease infectiousness (i.e., the probability of infection given a contact $\alpha$) (Fig 4).

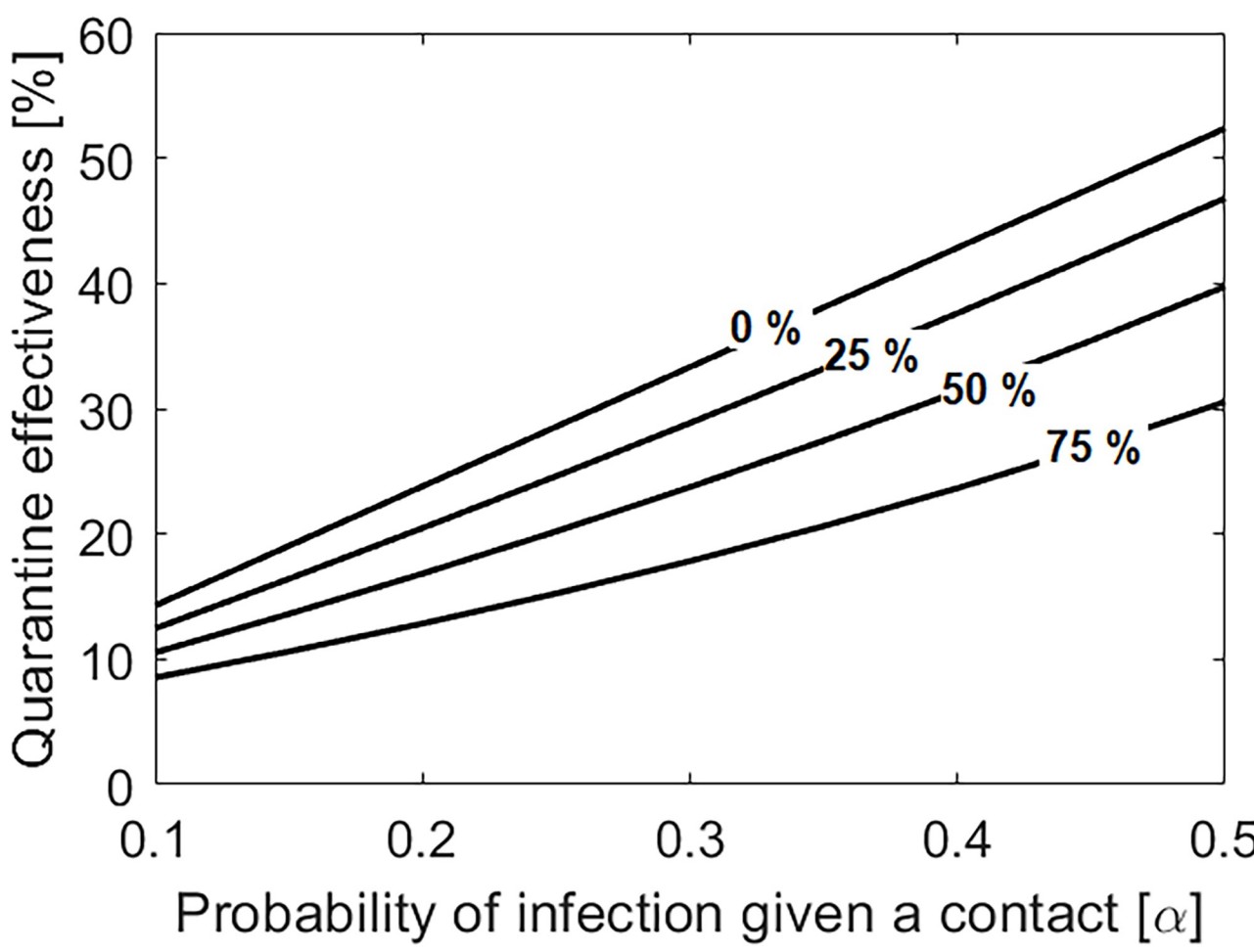

**Fig 4. Quarantine effectiveness, defined as the percentage of quarantined individuals that develop an infection, as a function of the probability of infection given a contact ($\alpha$).** Different curves represent different proportions of immune individuals in a population, where we consider that 0%, 25%, 50% or 75% of the population is immune.

Quarantine effectiveness decreases when the proportion of immune individuals in the population is high. Additionally, quarantine effectiveness increases when $\alpha$ is high, meaning that the percentage of quarantined individuals that develops an infection is higher in the presence of more contagious variants. For example, we obtain that for $\alpha = 0.2$, about 25% of the quarantined individuals will develop an infection in the absence of immune individuals. With 75% of the population being immune, only 10% of the quarantined individuals will develop an infection, thereby decreasing quarantine effectiveness by more than half. Note that quarantine effectiveness does not depend on the contact tracing efficiency $q$ (see Appendix A.6 in S1 File).

## 4 Discussion

Previous work has disputed whether contact tracing can be used as sole intervention to control outbreaks [6, 9, 30, 31]. Ferretti *et al.* [31] found that epidemic control through contact tracing could be achieved through the immediate notification and isolation of at least 70% of infectious cases, while three or more days delay in case notification would not allow for epidemic control. Analogously, we show that, under certain circumstances, efficient contact tracing alone can be considered an effective control measure even in nonimmune communities. For example, for a contact rate corresponding of 7 individuals per day, contact tracing can be an effective sole intervention as long as more than 50% of the contacts of symptomatic individuals are identified and quarantined within 1 or 2 days from symptoms onset. However, delays in detection and relaxation of border control measures can cause the number of contacts to be traced in a day to exceed the contact tracing capacity. Similarly, other studies found that testing at first symptom is a necessary prerequisite for efficient tracing [1, 2, 8, 10, 30], and that a higher contact tracing efficiency is needed to keep an outbreak under control when the number of initial cases is large [2, 32]. These findings emphasize the importance of testing at first symptoms, as well as testing new arrivals, to avoid overwhelming contact tracing capacity.

We find that investing in fast detection, for example via testing programs, rather than increasing the contact tracing capacity, has a larger impact in determining whether an outbreak will be controllable. Strong testing programs to ensure the quick detection of new community outbreaks, in combination with efficient identification and isolation of contacts, ensures slow epidemic spread, where the number of daily contacts to be traced remains low for the whole duration of the outbreak. Should slow detection cause uncontrolled epidemic spread, we expect overwhelming contact tracing capacity to occur even when the maximum daily number of tracing contacts is large, owing to exponential growth of the outbreak.

Population immunity has the double impact of reducing the contact tracing efficiency required to keep an outbreak under control, and minimizing the impact of delays and case importations. Indeed, in immune populations, a lower contact tracing efficiency is required to avoid overwhelming contact tracing capacity. For example, with 70% of the population being immune, a contact tracing efficiency of 40% is enough to keep an outbreak under control, even with a high contact rate of 7 individuals per day. Additionally, predictions show that the maximum number of contacts to be traced per day is drastically reduced when epidemic spread occurs in highly immune populations, where delays in detection or increase in the number of imported cases do not lead to a significant risk of overwhelming contact tracing capacity. These findings suggest possible downsizing of contact tracing practices in highly vaccinated communities or in communities whose populations have already experienced significant outbreaks, even when downsizing occurs in conjunction with the relaxation of social distancing and border restrictions. As immunity is distributed heterogeneously in the population, contact tracing downsizing, rather than dismantlement, should be considered, especially as contact tracing remains an important measure to reduce or avoid community spread in

communities that have not yet acquired immunity, such as schools for young children that may not be vaccine eligible.

Efficient tracing can be affected at many stages of the contact tracing process. Individuals may delay getting tested, and positive results may take days to be confirmed [33]. Additionally, contacts may not be easily identified or contacted, and they may not adhere to isolation requirements [10, 30, 33]. Generally, higher efficiency can be achieved in regions characterised by social cohesiveness, such as small jurisdictions with interconnected populations, where infected individuals might be known and a high proportion of contacts is likely to be reached [34, 35]. In denser populations, contact identification may be an arduous task, where manual contact tracing might be impractical and electronic contact tracing, for example through mobile apps, has often raised privacy concerns [36, 37]. Thus, while contact tracing might be an effective sole intervention in rural areas, failure might be observed in larger or more densely populated regions, which emphasizes the potential need for different policy decisions in small and large jurisdictions.

In our model, we assume contacts of symptomatic individuals to be isolated within 1–2 days, and we do not explicitly take into account possible delays from testing of symptomatic individuals to quarantining of their contacts. Additionally, we assume that individuals in quarantine do not transmit the disease, while this might often not be the case. Possible extensions of the model presented here include delays in the identification of contacts, and poor community adherence to quarantine rules [30]. Contact tracing could become more efficient by considering that pre-symptomatic and asymptomatic individuals, once identified as positive contacts of a symptomatic case, can as well contact trace. This particular feature could add realism to the model, but further complicate its formulation, and it will therefore be left for future work.

Finally, we show that quarantine effectiveness is low in highly immune populations, as a large proportion of quarantined contacts will not develop an infection. These findings suggest that a cost-effective approach is to establish different quarantine rules for immune and nonimmune individuals, as has indeed been done in several jurisdictions [38]. Rules should be evaluated with respect to the presence of more transmissible viral variants, which can increase the probability of infection given a contact for unvaccinated individuals as well as for individuals that have recovered from natural infection [11, 12]. Future modelling efforts should explicitly consider the risk of non-quarantining individuals that are only partially immune to different viral variants.

## Supporting information

**S1 File.** A, model description [16, 42, 43], [Table 1]. B, Contact tracing efficiency and controllable outbreaks.
(PDF)

## Author Contributions

**Conceptualization:** Maria M. Martignoni, Josh Renault, Joseph Baafi, Amy Hurford.

**Formal analysis:** Maria M. Martignoni, Josh Renault.

**Funding acquisition:** Amy Hurford.

**Investigation:** Maria M. Martignoni, Josh Renault, Joseph Baafi, Amy Hurford.

**Methodology:** Maria M. Martignoni, Josh Renault, Joseph Baafi, Amy Hurford.

**Supervision:** Amy Hurford.

**Validation:** Maria M. Martignoni, Amy Hurford.

**Visualization:** Maria M. Martignoni, Joseph Baafi.

**Writing – original draft:** Maria M. Martignoni, Josh Renault.

**Writing – review & editing:** Maria M. Martignoni, Amy Hurford.

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
