## [Decision Letter · Decision Letter 0]

3 Jan 2022

PONE-D-21-34590Downsizing of contact tracing during COVID-19 vaccine roll-outPLOS ONE

Dear Dr. Martignoni,

Thank you for submitting your manuscript to PLOS ONE. After careful consideration, we feel that it has merit but does not fully meet PLOS ONE’s publication criteria as it currently stands. Therefore, we invite you to submit a revised version of the manuscript that addresses the points raised during the review process.

Both reviewers have recommended minor changes which I would kindly ask you to follow in reviewing your manuscript.

We look forward to receiving your revised manuscript.

Kind regards,

Maria Vittoria Barbarossa, Ph.D.

Academic Editor

PLOS ONE

Journal Requirements:

“JR is supported by a National Sciences and Engineering Research Council of Canada (NSERC) Undergraduate Student Research Award (USRA). AH acknowledges financial support from an NSERC Discovery Grant, RGPIN 2014-05413. MM and AH are supported by Canadian Network for Modelling Infectious Diseases - Reseau canadien de modelisation des maladies infectieuses (CANMOD) and the Department of Health and Community Services, Government of Newfoundland and Labrador. AH acknowledges further support from the NSERC Emerging Infectious Disease Modelling Consortium. AH and JB are supported by the Atlantic Association for Research in the Mathematical Sciences and the New Brunswick Health Research Foundation.”

“JR is supported by a National Sciences and Engineering Research Council of Canada (NSERC) Undergraduate Student Research Award(USRA). AH acknowledges financial support from an NSERC Discovery Grant, RGPIN 2014-05413. MM and AH are supported by Canadian Network for Modelling Infectious Diseases - R´eseau canadien de mod´elisation des maladies infectieuses CANMOD) and the Department of Health and Community Services, Government of Newfoundland and Labrador. AH acknowledges further support from the NSERC Emerging Infectious Disease Modelling Consortium. AH and JB are supported by the Atlantic Association for Research in the Mathematical Sciences and the New Brunswick Health Research Foundation.”

We note that you have provided information within the Acknowledgements Section. Please note that funding information should not appear in the Acknowledgments section or other areas of your manuscript. We will only publish funding information present in the Funding Statement section of the online submission form.

“JR is supported by a National Sciences and Engineering Research Council of Canada (NSERC) Undergraduate Student Research Award (USRA). AH acknowledges financial support from an NSERC Discovery Grant, RGPIN 2014-05413. MM and AH are supported by Canadian Network for Modelling Infectious Diseases - Reseau canadien de modelisation des maladies infectieuses (CANMOD) and the Department of Health and Community Services, Government of Newfoundland and Labrador. AH acknowledges further support from the NSERC Emerging Infectious Disease Modelling Consortium. AH and JB are supported by the Atlantic Association for Research in the Mathematical Sciences and the New Brunswick Health Research Foundation.”

5. Please note that supplementary (should remain/ be uploaded) as separate "supporting information" files.

Reviewers' comments:

Reviewer's Responses to Questions

**Comments to the Author**

1. Is the manuscript technically sound, and do the data support the conclusions?

Reviewer #1: Yes

Reviewer #2: Yes

2. Has the statistical analysis been performed appropriately and rigorously? 

Reviewer #1: N/A

Reviewer #2: N/A

3. Have the authors made all data underlying the findings in their manuscript fully available?

Reviewer #1: Yes

Reviewer #2: Yes

4. Is the manuscript presented in an intelligible fashion and written in standard English?

Reviewer #1: Yes

Reviewer #2: Yes

5. Review Comments to the Author

Reviewer #1: In this paper, the authors establish a system of delay differential equations to model contact tracing during the COVID pandemic and study how vaccine roll-out and the relaxation of social distancing requirements affect contact tracing practises. During the beginning of a newly arising pandemic, contact tracing and quarantine are among the most straightforward and applicable intervention measures.

The model established in this work seems novel to me, at least I have not encountered a similar model for contact tracing yet. However, the main idea, namely considering transmission rates as a product of the average number of contacts is similar as done by Lipsitch et al. in [1], later followed e.g. by [2,3,4]. Up to my knowledge, it was the work by Lipsitch et al. where quarantine was considered this way in a compartmental model. I suggest the authors cite this paper or maybe also some of the ones using a similar method. An important difference between the present work and that of Lipsitch et al. is that in this paper, instead of contact with infected individuals, people are quarantined based on contact with contact traced people who are among those recently developing symptoms, i.e. among those who contracted the disease 5 days earlier. In the model, this is introduced correctly and (up to my knowledge) in a novel way. The authors then study the effect of various factors related the epidemic. However, the findings are not validated by application to real world situations. Some of the parameter values in Table 1 could be supported by available dta.

A couple of minor issues:

- To follow the typical structure of papers in Plos One, I suggest the authors to put the model and calculations in an appendix or supplementary material.

- Formula (11) for the cumulative number of quarantined is not clear to me, please clarify.

The manuscript may be checked to correct sime minor issues. Some examples:

- "an analytical criteria" in the Abstract

- p.9, l.-5: When -> when

[1] Lipsitch, M., Cohen, T., Cooper, B., Robins, J. M., Ma, S., James, L., et al. (2003). Transmission dynamics and control of severe acute respiratory syndrome.

Science, 300(5627), 1966e1970.

[2] Safi, M. A., & Gumel, A. B. (2010). Global asymptotic dynamics of a model for quarantine and isolation. Discrete and Continuous Dynamical Systems - Series B,

14(1), 209e231.

[3] Mubayi, A., Kribs Zaleta, C., Martcheva, M., & Castillo-Chavez, C. (2010). A cost-based comparison of quarantine strategies for new emerging diseases.

Mathematical Biosciences and Engineering, 7(3), 687e717.

[4] Barua, S., Dénes, A., Ibrahim, M. A., A seasonal model to assess intervention strategies for preventing periodic recurrence of Lassa fever, Heliyon 7(8):e07760(2021).

Reviewer #2: The paper "Downsizing of contact tracing during COVID-19 vaccine roll-out" investigates the interplay between vaccination and contact tracing in controlling the still ongoing COVID-19 pandemic.

The authors model the epidemic dynamics with a system of delay-differential equations which extends a previously published model by quarantine orders due to contact tracing and the effects of vaccination.

Within this model the authors, by solving the system of delayed differential equations numerically, explore in several scenarios what measures are necessary to avoid overwhelming the capacities of contact tracing (and thus a major outbreak).

The main messages of the paper are then that for contact tracing to be effective it has to be quick and efficient, vaccination supports contact tracing by reducing the efficiency of tracing necessary to keep the epidemic under control and that imported cases in populations with low vaccination quotas may quickly overwhelm contact tracing capacities.

The paper is well written and its methods and derivations are easy to understand and follow. The findings of the paper are largely supported by the evidence that the authors present. I recommend that the paper is accepted after the authors address the following points.

I have attached a .pdf file that contains my major and minor remarks.

6. PLOS authors have the option to publish the peer review history of their article (what does this mean?). If published, this will include your full peer review and any attached files.

Reviewer #1: No

Reviewer #2: No

---

## [Author Response · Author response to Decision Letter 0]

2 Mar 2022

The complete responses to reviewers comments are provided in the uploaded file named: 'response_to_reviewers'.

---

## [Decision Letter · Decision Letter 1]

3 May 2022

Downsizing of COVID-19 contact tracing in highly immune populations

PONE-D-21-34590R1

Dear Dr. Martignoni,

We’re pleased to inform you that your manuscript has been judged scientifically suitable for publication and will be formally accepted for publication once it meets all outstanding technical requirements.

Kind regards,

Maria Vittoria Barbarossa, Ph.D.

Academic Editor

PLOS ONE

Additional Editor Comments (optional):

I would like to apologize for the delay in sending out this acceptance notice. I had personal issues and was some offline for some time.

Reviewers' comments:

Reviewer's Responses to Questions

**Comments to the Author**

1. If the authors have adequately addressed your comments raised in a previous round of review and you feel that this manuscript is now acceptable for publication, you may indicate that here to bypass the “Comments to the Author” section, enter your conflict of interest statement in the “Confidential to Editor” section, and submit your "Accept" recommendation.

Reviewer #1: All comments have been addressed

Reviewer #2: (No Response)

2. Is the manuscript technically sound, and do the data support the conclusions?

Reviewer #1: Yes

Reviewer #2: Yes

3. Has the statistical analysis been performed appropriately and rigorously? 

Reviewer #1: N/A

Reviewer #2: Yes

4. Have the authors made all data underlying the findings in their manuscript fully available?

Reviewer #1: Yes

Reviewer #2: Yes

5. Is the manuscript presented in an intelligible fashion and written in standard English?

Reviewer #1: Yes

Reviewer #2: Yes

6. Review Comments to the Author

Reviewer #1: The authors have revised their manuscript according to the referees' reports. I suggest the manuscript to be accepted for publication.

Reviewer #2: I thank the authors of this manuscript for addressing the concerns that the second reviewer and me have pointed out. Most of the issues I have raised have been addressed with the first revision.

The authors have adapted their paper to a format conforming with the journals guidelines. In addition the MATLAB code used to generate the figures and simulations has been made public which I appreciate.

What follows are some minor remarks (mostly typos and phrasing) in no particular order that came to my attention while going through the revised manuscript. I recommend that the manuscript is accepted after these are addressed; as these are only very minute issues I do not require a second round of revisions.

1. Refercence 38 seems to have an issue with the author (Centres for Disease Control, Prevention, et al.)

2. Line 128: [...] S_0/N represents the proportion of immune individuals [...] should read [...] S_0/N represents the proportion of **initially susceptible** individuals [...]

3. Eq (2) still has p_c instead of p_{I_c}

4. Line 136 contains a \\tilde d which should probably be a \\tilde \\delta

7. PLOS authors have the option to publish the peer review history of their article (what does this mean?). If published, this will include your full peer review and any attached files.

Reviewer #1: No

Reviewer #2: No

---

## [Editor Report · Acceptance letter]

23 May 2022

PONE-D-21-34590R1 

Downsizing of COVID-19 contact tracing in highly immune populations 

Dear Dr. Martignoni:

I'm pleased to inform you that your manuscript has been deemed suitable for publication in PLOS ONE. Congratulations! Your manuscript is now with our production department. 

Kind regards, 

on behalf of

Dr. Maria Vittoria Barbarossa 

Academic Editor

PLOS ONE